# Transgenerational Epigenetic and Phenotypic Inheritance Across Five Generations in Sheep

**DOI:** 10.3390/ijms26136412

**Published:** 2025-07-03

**Authors:** Mehmet Kizilaslan, Camila U. Braz, Jessica Townsend, Todd Taylor, Thomas D. Crenshaw, Hasan Khatib

**Affiliations:** 1Department of Animal and Dairy Sciences, University of Wisconsin-Madison, Madison, WI 53706, USA; kizilaslan@wisc.edu (M.K.); jetownsend@wisc.edu (J.T.); toddtaylor@wisc.edu (T.T.); tdcrensh@wisc.edu (T.D.C.); 2Department of Animal Sciences, University of Illinois at Urbana-Champaign, Urbana, IL 61801, USA; cbraz@illinois.edu

**Keywords:** transgenerational epigenetic inheritance, paternal nutrition, environmental epigenetics, methionine, sheep, DNA methylation

## Abstract

Despite two extensive reprogramming events during early embryogenesis and gametogenesis, epigenetic information can be passed to the next generations, which constitutes the transgenerational epigenetic inheritance of phenotypes. Considering its utmost importance, there have been few studies focused on the transgenerational effects of dietary interventions, such as methionine supplementation, in livestock. Using whole-genome bisulfite sequencing, we implemented a single-base resolution differential methylation analysis for the F3 and F4 descendants of control vs. methionine-supplemented F0 twin-pair rams. Based on the results of our previous study on F0, F1, and F2 generations, we compared current results of 2981 and 1726 differentially methylated cytosines (DMCs), as well as 798 and 553 unique differentially methylated genes (DMGs), in F3 and F4, respectively. We identified 41 DMGs that exhibited transgenerational epigenetic inheritance (TEI-DMGs) across four generations and 11 TEI-DMGs across five generations. Finally, we estimated the effect size of F0 diet group on F3 and F4 growth and fertility-related phenotypes, providing evidence for transgenerational effects of diet group accompanying inherited differentially methylated genes. Here, for the first time using gene-level and phenotypic data, we demonstrate that a moderate dietary intervention can exert long-lasting transgenerational effects on offspring phenotypes extending beyond the F2 generation in sheep.

## 1. Introduction

Environmental epigenetics plays a pivotal role in shaping the expression patterns of genes beyond DNA sequence differences, influencing the phenotypic outcomes of organisms across generations [1]. Traditionally, the focus of the inheritance for complex traits has been on genetic alterations. However, environmental factors such as pollutants, stress, and dietary components can stimulate epigenetic modifications, including DNA methylation, histone modifications, and non-coding RNA regulation, which may persist through intergenerational and transgenerational inheritance [2]. While parental exposures directly influence intergenerational epigenetic effects, transgenerational inheritance extends beyond direct exposure, suggesting stable epigenetic reprogramming in the germline [3]. Intergenerational epigenetic inheritance refers to the inheritance of epigenetic marks and phenotypes by generations that are directly exposed to an environmental stimulus. In contrast, transgenerational epigenetic inheritance (TEI) describes the transmission of epigenetic and phenotypic changes to subsequent generations that were not directly exposed to the original environmental stimulus. These mechanisms challenge the traditional Mendelian inheritance with inferences that complex traits are influenced not only by genetic variation but also by epigenetic variation acquired from environmental exposure and inherited across generations. Some of the early examples of environmental epigenetic effects, although mostly focused on the phenotypic aspects of ancestral malnutrition, have been traced in survivors of the ‘Dutch Hunger Winter’ and the ‘Chinese Famine’ in humans [4,5]. Another example is the inherited epigenetic pattern across generations, accompanied by different coat colors (yellow vs. agouti) of the genetically identical ‘agouti’ mice [6].

Despite the two major epigenetic reprogramming events that occur during early embryogenesis and gametogenesis, certain epigenetic marks can evade erasure and contribute to the TEI of complex traits through the germline [7,8]. This has emerged as a compelling mechanism through which temporary environmental stimuli, including dietary interventions, can reshape the epigenome across multiple generations. Although the concept of TEI has been extensively documented in plants, fruit flies, and nematodes, there is limited evidence in mammals due to several factors extensively reviewed by Khatib et al. [3]. Accordingly, to overcome the lack of consensus and provide standards for TEI studies, five criteria were suggested as the TEI requirements. These are the simultaneous inheritance of phenotypes, epigenetic marks, and epimutations through unexposed generations, accompanied by germline inheritance and changes in gene expression [3]. Among the 20 examined studies focused on parental nutrition in mammals, none provided evidence for all five criteria, while 9 showed phenotypic differences, 3 demonstrated germline transmission, and only 1 provided epigenetic marks through the first unexposed generation, accompanied by associated phenotypic differences [3,8].

DNA methylation is an epigenetic mark that can be established de novo in response to environmental stimuli, maintained through mitosis, and inherited through both replicative and reconstructive mechanisms during meiosis in gametogenesis. It is then interpreted by transcription factors and DNA-binding proteins to regulate gene expression [9,10,11]. DNA methylation changes established in gametes—particularly in sperm and oocytes—are of critical importance, as these patterns can be transmitted to the next generation and may play a regulatory role in gene expression and phenotypic outcomes [12].

Recent advances in next-generation sequencing technologies enabled high-throughput screening for genome-wide DNA methylation landscape at a single base resolution level, using technologies such as methylation arrays, reduced representation bisulfite sequencing (RRBS), and whole-genome bisulfite sequencing (WGBS) [13]. DNA methylation is often the most proposed and investigated mechanism for transgenerational epigenetic inheritance in vertebrate phenotypes; changes are often linked to ancestral environmental exposures [14]. As previously noted, there is substantial evidence supporting the intergenerational epigenetic inheritance of DNA methylation patterns. However, studies specifically investigating TEI induced by environmental exposures in mammals remain limited, controversial, and supported by relatively scarce evidence [15,16,17,18,19]. While many studies examine parent-to-offspring transmission [3], definitive evidence of phenotype inheritance as well as epigenetic patterns beyond the germline of the first unexposed generation remains rare in mammals [3,11]. Agouti mice have been shown to have their coat color affected by maternal supplementation of methyl donors, but this effect is only transmitted over two generations and is lost by the F3 generation [20,21]. On the other hand, the detrimental effects of perinatal exposure to Bisphenol A were observed to persist into the F3 generation in fertility traits of male offspring in rats [22]. Another study imposed maternal nutrient restriction on F0 rats and traced the effects on blood pressure across F1, F2, and F3 [23]. However, these studies reported only phenotypic observations, without identifying specific epigenetic marks to account for the observed differences. In another study, the effect of the agricultural fungicide vinclozolin was observed in pregnant female rats [24]. Epigenetic changes were reported only in the F3 generation, and no corresponding phenotypic observations were documented, making it difficult to establish a clear link between the epigenetic modifications and the transmission of specific traits. Despite growing interest in TEI, studies investigating the influence of paternal nutrition remain exceptionally limited. To date, only a single study reported by our group has explored the transgenerational effects of paternal nutrition in sheep [8].

Methionine plays a central role in DNA methylation through its conversion into S-adenosylmethionine (SAM), the primary methyl group donor in the cell [25,26]. Previously, we demonstrated that methionine supplementation induced numerous intergenerational and transgenerational changes in the sperm DNA methylation patterns of sheep [8,25]. Sheep have emerged as an ideal model for studying epigenetic differences, owing to their short generation intervals, high rates of twinning, and the availability of a robust, well-annotated reference genome. Therefore, for our previous studies, we developed a ‘twin-pair methionine supplemented’ sheep model to study epigenetic differences in large animals [8,25]. In one particular study, exposure of F0 rams to paternal methionine supplementation led to alterations in the sperm DNA methylome and phenotypic changes in sheep that persisted across two subsequent generations (F1 and F2). Over 100 differentially methylated cytosines (DMCs) were inherited in the first unexposed generation, F2, showing correlations with gene expression and accompanying phenotypic differences [8]. The limited evidence available in mammals, combined with the promising results of our previous study, prompted us to continue tracking the same population to investigate the effects of methionine supplementation in F0 rams on sperm DNA methylation and phenotypes of F3 and F4 generations.

## 2. Results

### 2.1. Transgenerational Effects of Methionine Supplementation on Phenotypes

To observe the transgenerational effects of F0 paternal methionine supplementation on the phenotypes of F3 and F4 lambs, we implemented linear model estimations of the effect size on the available phenotypic measurements of F3 and F4 lambs (Table 1). Ultrasound loin muscle depth (LMD) and scrotal circumference (SC) were measured in 81 F3 and 95 F4 male lambs. Therefore, their analysis was performed in a sex-limited manner. LMD showed a significant association (*p* = 0.015) with the ancestral diet groups in F3 male lambs with control descendants having 1.56 mm greater loin muscle depth, consistent with the significant findings in F2 generation [8]. Additionally, a notable significance for SC was observed in pubertal males (*p* = 0.079). Specifically, males in the control group had on average, a 0.76 cm larger SC compared to the F3 descendants of methionine-treated F0 rams. The effect of F0 diet on birth weight (BWT) was also significant in F3 lambs (*p* = 0.026), with control lambs born 0.22 kg heavier than the treatment descendants. Weaning weight (WWT) of F3 lambs (adjusted 60 days) showed a notable significance in diet groups (*p* = 0.052), with control lambs having 1.02 kg lower weight compared to the treatment descendants. Finally, post-weaning weight (PWT), adjusted to 100 days after weaning, showed a notable significance (*p* = 0.081) with control descendants weighing 2.72 kg less than treatment descendants.

Linear modeling and significance testing for the effect of F0 diet group on F4 generation were performed for the body weights, LMD, and SC at puberty. Interestingly, an analysis of BWT revealed a reversed effect size of 0.23 kg in favor of methionine-supplemented sire descendants of F4 lambs (*p* = 0.033). Weaning weight showed the same trend as F3 and an effect size of 2.30 kg (*p* = 0.001) in favor of F4 descendants of methionine-fed F0 rams. The same trend was kept for 100 days after weaning with PWT being significantly affected by F0 diet groups (*p* = 0.001), with an effect size of 2.94 kg less PWT in the control group. Finally, an analysis of LMD and SC showed no significance of F0 diet group effects on F4 lambs (Table 1).

### 2.2. Overlapping and De Novo Differentially Methylated Cytosines (DMCs) Between Treatment vs. Control in F3 and F4

To identify the transgenerational effects of methionine supplementation in F0 rams, we employed WGBS and differential methylation analysis on the sperm DNA methylome of treatment versus control F3 and F4 rams. Accordingly, there were 2552 (1265 hypo- and 1287 hypermethylated) CpG-DMCs (Figure 1a), 97 (49 hypo- and 48 hypermethylated) CHG-DMCs (Figure 1c), and 332 CHH-DMCs (149 hypo- and 183 hypermethylated) (Figure 1e) observed between treatment and control groups in F3 rams. Additionally, 1495 (727 hypo- and 768 hypermethylated) CpG-DMCs (Figure 1b), 49 (28 hypo- and 21 hypermethylated) CHG-DMCs (Figure 1d), and 182 (95 hypo- and 87 hypermethylated) CHH-DMCs (Figure 1f) were observed in the F4 comparisons. To determine whether the F4 rams inherited any of the F3 DMCs, we examined the overlap of DMCs between the two generations. A total of 16 CpG-DMCs were found to be shared between F3 and F4 (Table 2). All identified DMCs as well as those overlapping between generations were provided (Appendix A).

To investigate if any of these DMCs were transgenerationally inherited (TEI-DMCs) from the ancestral population of F0, F1, and F2 rams [8], we checked for overlaps between these generations and our results for F3 and F4. Previously, we reported 107 overlapping DMCs (82 CpG-, 20 CHG-, and 5 CHH-DMCs) among the F0, F1, and F2 generations [8]. In the current study, no TEI-DMC overlaps were observed to persist beyond the F2 generation. However, F3 rams had only one DMC in common with F0, another one with F1, and eight DMCs with F2 rams, while F4 rams had only one DMC overlapping with F2 (Table 2). Interestingly, a gradual attenuation of methylation differences between the treatment and control groups was observed across the F0 to F4 generations.

To further investigate the genomic functions of these predominantly de novo methylation differences between treatment and control descendants, DMCs identified in the F3 and F4 generations were annotated based on their genomic locations (i.e., promoter, exon, intron, intergenic regions). Approximately 63% of all F3 and F4 DMCs were located in intergenic regions, followed by 36% in introns, with the remaining 1% in exons and promoters (Figure 2). These findings were generally consistent with observations in the F0, F1, and F2 generations, where approximately 65% of DMCs were located in intergenic regions, 32% in introns, and the remaining sites primarily in promoters, with a smaller proportion in exons [8].

### 2.3. Differentially Methylated Genes (DMGs) Between Treatment vs. Control in F3 and F4 Generations

To gain deeper insight into the biological functions associated with the observed methylation differences between treatment and control groups in the F3 and F4 generations, all DMCs were assigned to genes using the ‘Oar_rambouillet_v1.0’ reference genome assembly. Accordingly, 798 unique DMGs were identified between the treatment and control F3 rams. Top hypermethylated genes include *TTC22*, *ARHGAP29*, and *NTNG1*, while hypomethylated genes include *ZFYVE9*, *LDLRAD1*, and *DNAJC6*. Furthermore, 553 unique DMGs were observed between the two groups of F4 generation (Table 3). Among the top hypermethylated genes were *GPR33*, *SETD3*, and *EGFR*, while notable hypomethylated genes included *C6*, *MROH2B*, *SNORD123*, and *LARP1B* (Appendix A).

A clear dilution effect was observed, marked by a gradual decrease in the number of DMGs across successive generations (Table 3). Between the F3 and F4 generations, 114 overlapping DMGs were identified, most of which showed differential methylation primarily within intronic regions.

### 2.4. Transgenerationally Inherited Differentially Methylated Genes (TEI-DMGs)

Persistent phenotypic differences were observed between the control group and descendants of methionine-supplemented F0 rams across generations beyond F2. To investigate whether the apparent loss of transgenerationally inherited DMCs beyond F2 reflects a true loss of epigenetic inheritance or the emergence of compensatory mechanisms, we conducted a gene-level overlap analysis across all generations (Table 3; Appendix A). Remarkably, despite the limited overlap in individual DMCs between the F3 and F4 generations compared to the first three generations, a substantial number of DMGs exhibited consistent differential methylation across multiple generations. These findings suggest that, although the exact same DMCs within individual genes may not be consistently inherited or detected across generations, gene-level differential methylation can persist as a transgenerational response to F0 paternal methionine supplementation. We identified 41 TEI-DMGs consistently differentially methylated across the first four consecutive generations (i.e., F0, F1, F2, and F3). Notably, 11 of these TEI-DMGs remained differentially methylated through the five generations (i.e., F0, F1, F2, F3, and F4). A great share of these TEI-DMGs (~90%) were recursively methylated at multiple intronic cytosines. These five generations spanning 11 TEI-DMGs include coiled-coil domain-containing 171 (*CCDC171*), Rho GTPase activating protein 15 (*ARHGAP15*), low density lipoprotein receptor-related protein 1B (*LRP1B*), thrombospondin type I domain-containing 7A (*THSD7A*), contactin-associated protein 2 (*CNTNAP2*), adhesion G protein-coupled receptor B3 (*ADGRB3*), protocadherin 9 (*PCDH9*), discoidin domain receptor tyrosine kinase 2 (*DDR2*), roundabout axon guidance receptor homolog 1 (*ROBO1*), dipeptidyl peptidase like 10 (*DPP10*), and Usher syndrome 2A (*USH2A*). Among these genes, four (*CCDC171*, *ARHGAP15*, *LRP1B*, *DPP10*) were located on chromosome 2, two (*THSD7A*, *CNTNAP2*) were on chromosome 4, two others (*DDR2*, *ROBO1*) were on chromosome 1, and the others were chromosome 9 (*ADGRB3*), 10 (*PCDH9*), and 12 (*USH2A*). These genes are listed in Appendix A.

### 2.5. Functional Annotation of DMGs and TEI-DMGs

To understand the biological relevance of DMGs in F3 and F4, we performed an annotation enrichment analysis to identify associated biological processes and the KEGG pathways in which these genes are involved (Figure 3). Accordingly, most of the F3-DMGs were associated with biological processes, including cellular morphogenesis, synaptic signaling, neuronal development, and developmental growth, with predominantly molecular functions such as ion transmembrane transporter activity, calcium ion binding, and ion channel activity (Figure 3a). Additionally, KEGG biological pathways, including axon guidance, glutamatergic synapses, phosphatidylinositol signaling, and TGF-β signaling, were significantly enriched among the F3-DMGs (Figure 3b). The same pattern is largely conserved by the F4-DMGs, which primarily involve cellular morphogenesis, neuron development, and synapse organization, with some exhibiting molecular functions related to GTPase regulation and ligand-gated ion channel activities (Figure 3c). No KEGG pathways were identified for F4-DMGs to be annotated.

To better comprehend the biological context of the gene-level transgenerational inheritance among the generations, we further enriched the annotation of 41 identified TEI-DMGs among F0, F1, F2, and F3 (Figure 4a) and 11 for all five generations, namely F0, F1, F2, F3, and F4 (Figure 4b). Interestingly, a consistent enrichment of functions in cell morphogenesis and neuron development was observed in all comparisons across generations.

## 3. Discussion

In our previous work, we demonstrated that environmentally induced epigenetic marks in F0 showed TEI in sheep [8]. We identified 107 TEI-DMCs shared across the F0, F1, and F2 generations, accompanied by growth- and fertility-related phenotypic differences and downstream effects on gene expression, in response to methionine supplementation of F0 rams only [8]. In the current study, we extended our investigation to the F3 and F4 generations of the same population to assess the long-term consequences and transgenerational effects of ancestral paternal methionine supplementation on the sperm DNA methylome, as well as growth- and fertility-related traits in these later generations. Our study recontextualizes the transgenerational impact of dietary interventions by presenting a two-dimensional perspective—DMC and DMG levels—through single-base resolution analysis. For the first time, we provide compelling evidence that DMGs can be epigenetically inherited, along with distinct phenotypic patterns, across five consecutive generations (F0, F1, F2, F3, and F4). Remarkably, these transgenerational effects were triggered solely in response to a 12-week prepubertal paternal methionine supplementation in the F0 rams, emphasizing the enduring epigenetic legacy of a single moderate dietary exposure.

In our study, growth- and development-related traits, such as birth, weaning, and post-weaning weights, as well as loin muscle depth and the fertility-related trait scrotal circumference, showed statistically significant associations with the F0 dietary treatment in F3 and F4 generations. In our previous study, scrotal circumference was significantly affected by the F0 diet in both F1 and F2 generations [8]. Our current findings extend this effect to the F3 generation; however, the association was no longer significant in F4 rams. A similar pattern was observed for loin muscle depth, which was associated with the F0 diet in F2 (previous study) and F3 (current study), but the effect is lost in F4. On the other hand, birth weight, although not provided in the previous study, was significantly influenced by the F0 diet in both F3 and F4 generations. In the F3 generation, birth weight was higher in the control group compared to the treatment group; however, this effect was reversed in the F4 generation. The mechanisms underlying these generational reversals are not yet fully understood. A negative correlation between direct and maternal genetic effects has been reported in multiple livestock studies. For example, genes associated with increased body weight in the dam may restrict fetal growth in her offspring [27]. Additionally, DNA methylation marks linked to specific phenotypes are reversible across generations, as observed in our study and others [8]. It is also possible that de novo methylated cytosines contributed to the observed reversal of phenotypic effects across generations. Weaning and post-weaning weights were influenced by the F0 diet in F3 and F4 generations, consistent with the effects observed in F1 and F2 in the previous study.

A substantial number of de novo methylation changes were observed at specific loci in the F3 and F4 generations when comparing descendants of methionine-supplemented rams to those of control rams, despite the original treatment being applied five generations earlier (i.e., F0). Using WGBS at a single-base resolution, we identified 2552 CpG-DMCs, 97 CHG-DMCs, and 332 CHH-DMCs in the sperm of F3 rams that were differentially methylated between the two groups of F0 ram descendants. In the F4 generation, we detected 1495 CpG-DMCs, 49 CHG-DMCs, and 182 CHH-DMCs. Many of these DMCs were mapped to genes involved in cellular morphogenesis, neuron development, and developmental growth, which may help explain the phenotypic differences observed between treatment control descendants in F3 lambs. Our study uncovered various non-CpG methylation differences (CHG- and CHH-DMCs). Studies on non-CpG methylation suggest that it is highly conserved across species, dynamically regulated in a tissue-specific manner, and potentially involved in the evolution and regulation of transposable elements, supported by evidence from retrotransposon methylation and genomic stability [28,29]. An interesting finding from our study is the high abundance of DMCs located within gene body elements. While cytosine methylation in promoter regions is well established as a mechanism for suppressing gene expression—either by hindering transcription factor binding or by recruiting transcriptional repressors—methylation within gene bodies (i.e., exons and introns) exhibits a more variable influence on gene expression [30,31,32]. Intron methylation, specifically in the first intron, is suggested to increase gene expression [33,34]. Importantly, the methylation of introns has recently also been proposed to affect intron retention in mature mRNA, thereby contributing to alternative splicing and gene expression [35].

On the other hand, although a few generation-specific overlaps were observed in our study, we were unable to observe TEI-DMCs extending beyond the F2 generation. A similar pattern has been reported using a differentially methylated region (DMR)-based approach in gestating rats exposed to herbicides, where researchers similarly found de novo DMRs in each generation without any overlap across F1, F2, and F3 [36]. Moreover, Ben Maamar et al. reported that exposure of rats to vinclozolin during gestation resulted in 44 DMRs overlapping between F2 and F3, 4 between F1 and F3, and 2 between F1 and F2; however, none were shared across all three generations [37]. Several factors may contribute to the challenges faced in the identification process of transgenerationally persistent DMCs. These include multiple reprogramming events of the epigenome across generations, which reduces the likelihood of overlapping, short-term environmental influences that each generation experiences, dilution of epigenetic marks over time, and stochastic variation, as well as inconsistent sequencing coverage of the same loci across generations. Moreover, TEI involves a complex interplay among small non-coding RNAs (sncRNAs), DNA methylation, RNA methylation, and histone modifications. These mechanisms do not operate independently but interact dynamically to influence gene expression patterns across generations [11,38]. For instance, evidence suggests that sperm-derived miRNAs have been shown to mediate both DNA methylation and histone modifications following fertilization and during embryogenesis [39]. Therefore, the emergence of de novo DMCs and the lack of consistent DMC overlap across generations in our study may also reflect the influence of these interconnected epigenetic pathways.

The persistent phenotypic differences observed across generations, despite the lack of inherited DMCs in the F3 and F4 generations, prompted us to investigate DMGs rather than single methylated loci as an alternative dimension of epigenetic information that might account for these transgenerational phenotypes. Our results suggest that paternal methionine supplementation may have triggered a cascade of gene-level methylation changes that persisted across generations, albeit with a gradual attenuation of the signal over time. We identified 41 genes that were consistently differentially methylated across four generations (F0, F1, F2, and F3), which were subsequently reduced to 11 genes retaining differential methylation that persisted across sperm samples from all five generations (F0, F1, F2, F3, and F4). Notably, all 11 of these genes have been previously associated with growth- and development-related diseases and phenotypes in humans, including body weight, height, body mass index, and nervous system development [40]. For instance, *DDR2* has been implicated in bone growth and development in humans and mice [41]. Although *ROBO1* was previously known to be involved in brain development, recent studies have shown its pleiotropic effects on postnatal growth and lifespan in mice [42]. Interestingly, *LRP1B* has been repeatedly associated with body mass index and childhood obesity [43]. Additionally, *ARHGAP15* has been implicated in classical and endocrine fertility traits in Swedish Red and Holstein cattle [44]. Several other genes, including *CCD171*, *CNTNAP2*, *DPP10*, and *PCDH9* (AS3), have been reported to be differentially expressed during germ cell development and are associated with male fertility, according to the Male Fertility Gene Atlas [45]. Thus, our findings further support the transgenerational epigenetic effects of paternal methionine supplementation by identifying heritable phenotypic traits potentially influenced by this exposure and by providing evidence of TEI-associated DMGs. The mechanism we propose for TEI-associated DMGs may share certain similarities with the regulation of imprinted genes. Imprinted genes are controlled by differentially methylated imprinting control regions (ICRs), and many of these genes possess multiple ICRs, as documented in the Imprinted Gene Database (www.geneimprint.com). The precise methylation sites within ICRs can vary depending on the developmental stage, tissue type, and evolutionary context, while still maintaining the imprinted status of the gene [46,47,48]. These studies highlight that while a gene’s imprinted status may be preserved, the specific loci responsible for maintaining this imprinting can vary in their location. Consistent with this, our findings demonstrate that gene-level differential methylation patterns can persist across five generations, even if the specific DMCs involved change over time. However, the underlying mechanisms driving this phenomenon remain poorly understood and require further investigation.

A notable finding of our study is the enrichment of genes associated with nervous system development, neuron projection, synaptic signaling, and brain development at both the DMG and TEI-DMG levels. These results align with our previous study on the F0, F1, and F2 generations, which also reported enrichment of similar gene functions [8]. Additionally, another of our earlier studies examining the effects of paternal methionine supplementation on sperm DNA methylation and the embryonic transcriptome in sheep identified sperm-expressed genes involved in nervous system development [49]. Other studies have similarly reported striking overlaps in gene and protein expression patterns between the brain and testes in both mice and humans [50,51]. Emerging evidence increasingly supports a functional and molecular link between brain and sperm biology [52]. Thus, our findings suggest that the same set of genes influenced by ancestral paternal methionine supplementation in sheep sperm may also play pleiotropic roles in brain and nervous system development, underscoring the evolutionarily conserved relationship between these two systems in terms of molecular pathways and regulatory mechanisms.

One limitation of our study stems from the technical constraints of WGBS. Achieving consistent coverage of the same cytosine loci across all samples remains challenging, even at high sequencing depths. This variability can result in a reduced overlap of specific methylation sites across generations. Further investigation is needed to elucidate the molecular mechanisms underlying the maintenance and transmission of epigenetic information across tissues and generations. Understanding the prevalence of intergenerational and transgenerational epigenetic phenomena in natural systems—and their physiological significance in biological contexts—remains a critical area of research. The genes identified in this study represent strong candidates for future validation and functional studies. Approaches such as genome and epigenome editing could be employed to explore their potential roles in shaping the observed phenotypes.

## 4. Materials and Methods

All procedures involving animals were approved by the Institutional Animal Care and Use Committee of the University of Wisconsin-Madison (Protocol ID: A006488).

### 4.1. Experimental Design, Sample Collection, and Phenotypes

The experimental design of the study for the F0, F1, and F2 rams was previously described [8,25]. In brief, each of the 20 male Polypay twin pairs had one twin randomly assigned to the control diet (control group), while the other received the same diet supplemented with rumen-protected methionine (0.22% addition, 1.5 g; RPM Smartamine, Adisseo, Alpharetta, GA, USA) (treatment group) from weaning until puberty (12 weeks). Methionine was selected as the dietary stimulus due to its key role as a methyl donor in DNA methylation [26]. Among those, 5 twin pairs (F0 rams) were selected to mate 8–9 control diet-fed Polypay ewes each. Dietary intervention was limited to the F0 males, and all subsequent generations were fed the control diet and were raised under identical controlled conditions. The F1 generation comprises 225 lambs (i.e., 115 males and 110 females). To produce the F2 generation, 10 F1 rams (progeny of each F0 control and treatment sires) were mated to 10 Polypay ewes each. The F2 generation comprises 188 animals (94 males and 94 females). The same breeding strategy was applied to produce F3 and F4 generations, resulting in 178 F3 lambs (87 males and 91 females) and 224 F4 lambs (113 males and 111 females).

Sperm samples of the F3 and F4 rams were collected at puberty using electroejaculation via the Lane Pulsator IV (Lane Manufacturing Inc., Denver, CO, USA). Immediately after semen collection, each sample was diluted 1:1 with a prewarmed 37 °C semen extender. Ram puberty was assessed based on sperm motility and concentration using the iSperm (Aidmics Biotechnology Co., Taipei, Taiwan). Only ejaculates containing a minimum of 50 million sperm and demonstrating at least 10% motility were included in further analyses, as these thresholds are indicative of puberty [53]. The semen samples were washed with phosphate-buffered saline (PBS), centrifuged to form pellets, and stored in RNAlater at −80 °C until use.

To estimate the effect of the F0 methionine supplementation on growth and reproductive traits of the F3 and F4 generations, BWT, WWT, PWT, LMD, and SC were recorded in varying number of animals (Table 1). SC was measured at the widest point of the scrotum with a tape measure. Ultrasound measurements were carried out via an Aloka SSD-500 portable ultrasound machine with a 7.5 MHz linear probe, which was used to visualize the sheared left side of the F3 and F4 rams on the rib-eye muscle area between the 12th and the 13th ribs. The Image J software v1.53e was used to obtain LMD, measured at the vertically deepest point of the muscle.

Several covariates and environmental factors were also recorded to account for while fitting linear models to determine the significance of the fixed effect of the F0 diet group on the phenotypic observations of F3 and F4. These are sex, age in days at measurement (covariate), age of dam (covariate), birth type (single, twin, triplet, quadruplet), and rearing type (single, naturally, artificially, foster-reared). A generalized linear model provided by R (v4.4.2) environment was used to estimate the effect sizes, followed by a Wald test to identify the significance of the effect sizes [54]. For LMD and SC, the weight of the animal at measurement was also included as a covariate.

### 4.2. DNA Extraction and WGBS

In total, 10 F3 and 10 F4 sperm samples were selected for WGBS, with an equal representation of descendants from control and treatment F0 rams (*n* = 5 per diet group in each generation). Genomic DNA was extracted from sperm samples using the Quick-DNA Miniprep Plus (Zymo Research, Irvine, CA, US). WGBS was undertaken by the Genomics unit of the Roy J. Carver Biotechnology Center at the University of Illinois, Urbana-Champaign. Prior to sequencing, the DNA samples were bisulfite-treated using the Zymo EZ DNA Methylation-Lightning kit, followed by library preparation with Illumina universal adapter sequences and the Pico Methyl-Seq Library Prep Kit (Zymo Research). Samples were sequenced with the Illumina NovaSeq 6000 platform (Illumina, San Diego, CA, USA) on an S4 flow cell to generate ~30× mean coverage of 150 bp length paired-end reads. In total, approximately 5 billion paired-end reads were obtained for all samples, with a distribution ranging from 215 million to 293 million, and an average of 260 million across samples. ‘Fastq’ files were generated and demultiplexed using the ‘bcl2fastq’ v2.20 Conversion Software (Illumina) for downstream analysis.

### 4.3. WGBS Alignment

Following demultiplexing, ‘fastq’ files were subjected to quality control (QC) and trimming steps using FastQC v0.12.1 and Trim Galore v0.6.10 and incorporating Cutadapt v4.8 software to remove low quality bases, reads, and adapter sequences [55,56,57]. An ASCII +33 quality score was used as Phred score for trimming, and reads shorter than 30 bp were excluded from further analysis. High-quality reads were then non-directionally aligned to the ‘Oar_rambouillet_v1.0’ reference genome assembly obtained from the NCBI database using Bismark v0.24.2, a bisulfite-aware aligner indirectly using an FM index based on Burrows–Wheeler transform with Bowtie2 v2.5.4 software [58,59]. On average, the percentage of alignment was 75%, ranging from 65% to 80%. Following deduplication, methylation coverage files were obtained for cytosines on CpG islands, as well as in CHG and CHH contexts, where H is A, T, or C, using the ‘bismark_methylation_extractor’ function at a single base resolution. The reference genome ‘Oar_rambouillet_v1.0’ is particularly used for consistency in loci and gene annotations with previous work on F0, F1, and F2 generations [8] and in the current study. The Bismark coverage files obtained here were later transferred into the R environment to undertake any further analysis, including identification of DMCs and DMGs.

### 4.4. Differential Methylation Analysis

To identify the DMCs stimulated by methionine supplementation of F0 rams in sperm of F3 (5 treatment vs. 5 control descendants), as well as F4 (5 treatment vs. 5 control descendants) generation rams, we focused on comparing methylation counts of CpG, CHG, and CHH contexts of their sperm genome at a single base resolution. CpG, CHG, and CHH filtering, preliminary checks with Pearson’s correlation, hierarchical clustering, Principal Component Analysis (PCA) and the detection of differentially methylated cytosines (DMCs) were all conducted using the ‘methylKit’ v1.32.1 R package [60]. Cytosines were filtered to have a read count of >10 to reduce Type I and possible Type II error rates. Additionally, cytosines on sex chromosomes and mitochondrial DNA, as well as those with methylation levels of ≥99.9%, were also filtered out from further analysis. This was followed by median normalization of the read coverages between samples. With an initial inspection of Pearson’s correlation table, hierarchical clustering and PCA, 1 control sample from the F3 generation, as well as 1 treatment sample of the F4 generation, were excluded from further analysis due to extremely low intra- and intergroup correlation and concordances. Therefore, DMC analysis was implemented with 4 controls vs. 5 treatments in F3, while it was performed with 5 controls and 4 treatment rams in F4.

A beta-binomial regression—Bayesian hierarchical model—provided by ‘methylKit’ R package was used to detect methylation count differences in CpG, CHG, and CHH contexts between the treatment and control groups of F3 and F4 rams [60,61]. Beta-binomial models are known to robustly account for the overdispersion of variance in methylation counts, with an estimation of beta-binomial dispersion parameter, thus reducing the Type I error rate [62]. Briefly, a beta-binomial dispersion parameter obtained from a genome-wide prior distribution was used to estimate the distribution of methylation levels in each group of replicates. Then the means of these distributions were tested for equality with Wald’s F test. The model, parameter estimations, and statistical test procedures are detailed by [61]. *p*-values of the statistical tests were adjusted by a Benjamini–Hochberg procedure (FDR correction) to further suppress the false positive rate due to multiple testing [63]. Finally, two stringent significance criteria were applied: only cytosines exhibiting a methylation difference of ≥20% and a q-value ≤ 0.01 were classified as DMCs. DMCs with higher methylation levels in treatment animals compared to controls were classified as hypermethylated, whereas those with lower methylation levels were designated as hypomethylated. DMCs identified in the F3 and F4 groups were both compared with each other and with those previously reported in the F0, F1, and F2 generations by [8] to identify potential overlaps indicative of transgenerationally inherited methylation marks.

### 4.5. Calling Gene Components, DMGs, and TEI-DMGs

Following the detection of DMCs, the ‘Oar_rambouillet_v1.0’ reference genome annotations were used with the R packages ‘genomation’ v1.38.0 and ‘GenomicRanges’ v1.58.0 to assess the distribution of methylation differences across gene components (e.g., promoters, introns, exons, intergenic regions), as well as the corresponding gene names with their distances to the transcription start sites [64,65]. DMGs were defined as genes with a DMC located within the promoter and gene bodies (exon, intron) or the ±20 Kb distance to its transcription start site (TSS). The identified DMGs were also compared with those from our previous analyses of the F0, F1, and F2 generations to identify overlapping genes that may represent potential TEI-DMGs.

### 4.6. Functional Annotation of the DMGs and TEI-DMGs

After the identification of DMGs and TEI-DMGs, the obtained gene lists were subjected to functional enrichment analysis under the FDR < 0.05 criterion using ‘ShinyGO’ v0.82. The biological processes (BPs) in which the specified genes are involved were obtained, along with Gene Ontology (GO) terms. Kyoto Encyclopedia of Genes and Genomes (KEGG) pathways related to the identified genes were provided where possible. Additionally, the DAVID Bioinformatics Tool v2025_1 was used to annotate the identified genes with diseases, phenotypes, and further functional annotations in humans [66]. Since annotation is well established in human and mice genomes, orthology between the species was utilized for inferences to sheep.

## 5. Conclusions

Our results provide the first evidence of persistent transgenerational inheritance of differentially methylated genes (TEI-DMGs), de novo DMGs, and phenotypes across five generations, environmentally stimulated by F0 paternal methionine supplementation. The results of this study enhance our understanding of the interplay between environmental stimuli and epigenetic inheritance mechanisms. This study also suggests that paternal dietary interventions can induce de novo differentially methylated cytosines in sperm, persisting up to four generations beyond the directly exposed individuals, even in the absence of continued environmental exposure. We also demonstrated that focusing on differentially methylated cytosines (DMCs), rather than differentially methylated regions (DMRs), can help mitigate limitations associated with sequencing coverage variability by emphasizing gene-level patterns across generations. Most importantly, using a large animal model, we demonstrated that paternal lifestyle could influence growth, fertility, and health, particularly through nervous system-related genes, in descendants up to four generations via the germline. This finding introduces a novel framework for understanding the inheritance of complex traits and for developing preventative strategies for common diseases.

## Figures and Tables

**Figure 1 ijms-26-06412-f001:**
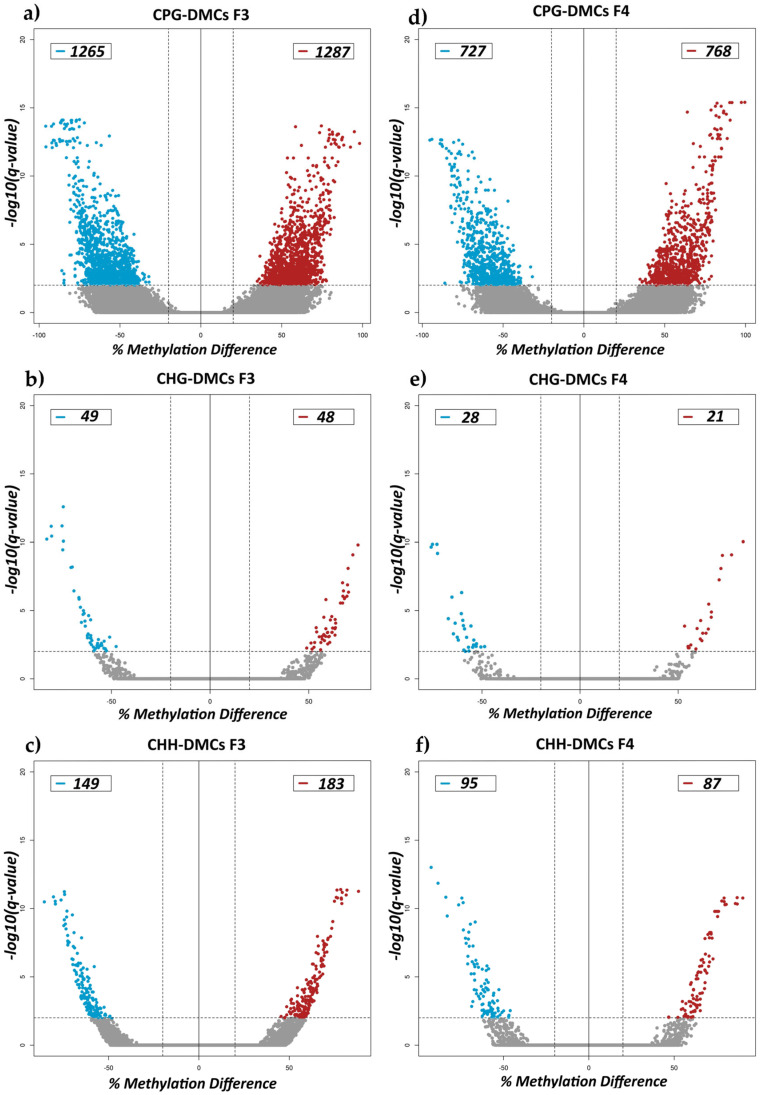
Volcano plots for differential methylation analysis of cytosines in CpG, CHG, and CHH contexts of F3 and F4 rams. Plots show the FDR corrected *p*-values (q-values) on −log10 scale on *y*-axis and % methylation difference in DMCs between the two groups on the *x*-axis. Dashed vertical lines represent ± 20% methylation difference thresholds, while the horizontal line marks the 0.01 q-value threshold. Finally, blue points represent hypomethylated DMCs, red points indicate hypermethylated DMCs, and gray points correspond to tested cytosines that were not statistically significant. The legend also displays the number of significant DMCs identified in each category. (**a**,**c**,**e**) show the % methylation difference and −log10(q-value) of significant DMCs in CpG, CHG, and CHH contexts of F3, respectively. (**b**,**d**,**f**) show the % methylation difference and −log10(q-value) of those significant DMCs in CpG, CHG, and CHH contexts of F4.

**Figure 2 ijms-26-06412-f002:**
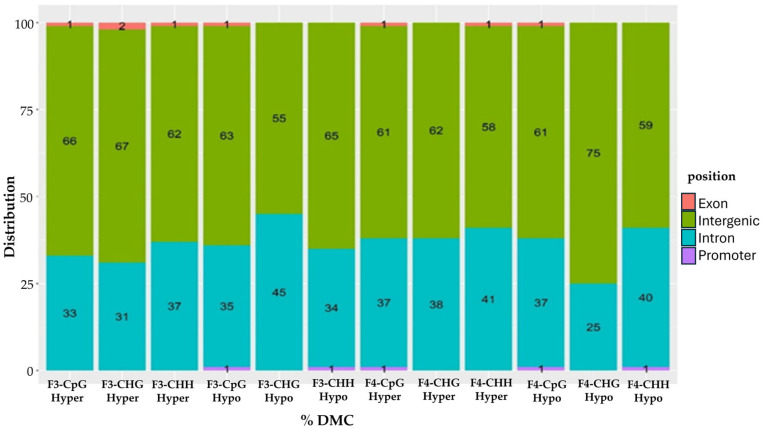
Distribution of genomic contexts for F3 and F4 DMCs. The numbers on the bars indicate the percentage of DMCs found within each color-coded genomic context, as defined in the legend (i.e., genomic position). Each bar represents the distribution of hyper- and hypomethylated DMCs across CpG, CHG, and CHH contexts in the F3 and F4 generations.

**Figure 3 ijms-26-06412-f003:**
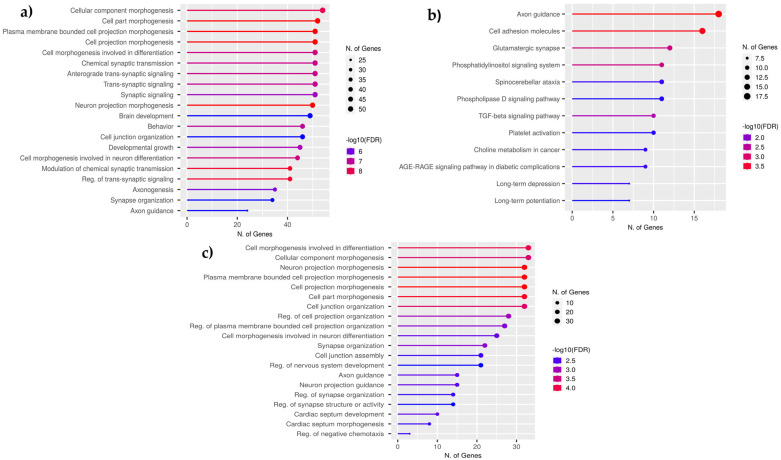
Functional enrichment plots of F3 and F4 DMGs are plotted based on the number of genes involved in each term and the −log10(FDR) of enrichment analysis. (**a**) Biological processes involved by the majority of the identified F3-DMGs. (**b**) KEGG pathways involved by the given number of F3-DMGs. (**c**) Biological processes involved by the majority of the identified F4-DMGs.

**Figure 4 ijms-26-06412-f004:**
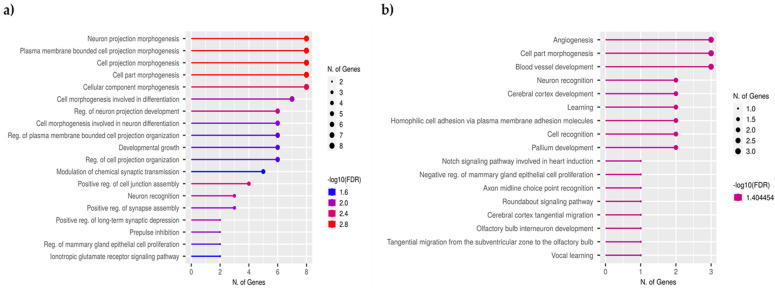
Functional enrichment plots of TEI-DMGs are plotted based on the number of genes involved in each term and the -log10(FDR) of enrichment analysis. (**a**) Biological processes involved by 41 TEI-DMGs observed across four generations (F0, F1, F2, and F3). (**b**) Biological processes involved by 11 TEI-DMGs were identified as common across five generations (F0, F1, F2, F3, and F4).

**Table 1 ijms-26-06412-t001:** Effect of F0 paternal methionine supplementation on F3 and F4 generation phenotypes.

	F3 Generation			F4 Generation		
Trait	N	Effect Size	*p*-Value	N	Effect Size	*p*-Value
BWT (kg)	178	−0.22	0.026	224	0.23	0.033
WWT (kg)	178	1.02	0.052	215	2.30	0.001
PWT (kg)	162	2.72	0.081	202	2.94	0.001
LMD (mm)	81	−1.56	0.015	95	NS	NS
SC (cm)	81	−0.76	0.079	95	NS	NS

**Notes:** BWT, birth weight; WWT, weaning weight; PWT, postweaning weight; SC, scrotal circumference; LMD, loin muscle depth; N, number of animals analyzed; Effect size, fixed effect size of F0 diet groups (control vs. treatment); NS, not significant.

**Table 2 ijms-26-06412-t002:** Number of differentially methylated cytosines in sheep sperm across generations.

Generation	F0	F1	F2	F3	F4
**F0**	7286				
**F1**	1007	3483			
**F2**	347	272	4630		
**F3**	1	1	8	2981	
**F4**	0	0	1	16	1726

**Notes:** Numbers include CPG-, CHG-, and CHH-DMCs observed across generations. Diagonal values show the total number of differentially methylated cytosines in the treatment vs. control comparisons of each generation, while values below diagonal are the overlap (i.e., same direction of methylation status) between the differentially methylated cytosines across generations.

**Table 3 ijms-26-06412-t003:** Number of differentially methylated genes (DMGs) in each generation and across generations, including transgenerationally inherited differentially methylated genes (TEI-DMGs).

Generation	DMGs
F0	1388
F1	817
F2	1069
F3	798
F4	553
F0-F1	414
F0-F1-F2	181
F0-F1-F2-F3	41
F0-F1-F2-F3-F4	11

## Data Availability

The data of the study will be available through NCBI GEO with accession number GSE301579 following publication.

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
