# Peer review of "Transgenerational Epigenetic and Phenotypic Inheritance Across Five Generations in Sheep"

_ijms, 2025, doi:10.3390/ijms26136412_

Round 1

Reviewer 1 Report

Comments and Suggestions for Authors

The study demonstrating diet-induced transgenerational effects across five generations in sheep is of significant scientific merit and deserves appreciation. However, I have several questions that require clarification:
1.In line 27 of the article, the term "first" is used. Can you guarantee this is the first instance of such a finding? If not, the term may lack rigor and could be reconsidered. 
2.The text states that sperm samples from 10 sheep each in the F3 and F4 generations were analyzed. Is this sample size too small? Please explain the rationale for choosing this number.
 3. The F3 generation showed higher birth weight (BWT) in the control group, while the F4 generation reversed this trend, with higher BWT in the methionine group. The article does not explain this phenomenon. Please provide an interpretation. 
4. In Figure 1, key gene loci for the F3/F4 generations are not clearly labeled. Please supplement the figure or provide an explanation.
 5. The distinction between "transgenerational epigenetic inheritance" and "intergenerational effects" is unclear. We suggest briefly explaining and standardizing the definitions of these terms.
 6. The discussion cites transgenerational studies in rats and mice but does not clarify the similarities and differences in epigenetic regulation between sheep and rodents. We recommend adding a species-specific analysis to avoid over-extrapolating conclusions.

Author Response

Reviewer 1 comments and suggestions:

The study demonstrating diet-induced transgenerational effects across five generations in sheep is of significant scientific merit and deserves appreciation. However, I have several questions that require clarification:

1. In line 27 of the article, the term "first" is used. Can you guarantee this is the first instance of such a finding? If not, the term may lack rigor and could be reconsidered.

Response: We appreciate the reviewer’s comment. Indeed, this is the first study to report the inheritance of both phenotypes and gene expression patterns across five generations, not only in sheep, but in any mammalian species. While many studies have demonstrated the inheritance of either phenotypes or epigenetic marks independently, few, if any, have shown the co-inheritance of both. This distinction is critical for advancing our understanding and definition of transgenerational epigenetic inheritance.

2. The text states that sperm samples from 10 sheep each in the F3 and F4 generations were analyzed. Is this sample size too small? Please explain the rationale for choosing this number.

Response: The method we used in our differential methylation analysis is a beta binomial distribution-based regression analysis (an empirical Bayes method) followed by an F-test that controls overdispersion. This method reduces false positive rates and increases statistical power (Feng et al., 2014: Feng and Wu, 2019). Additionally, we restrained our analysis to the regions above 10X coverage and aimed for at least 15% methylation difference, which further boosts the statistical power of our analysis. In our previous study we used similar parameters with the same number of animals and obtained hundreds to thousands of differentially methylated loci (Braz et al., 2022). Finally, we also used the descendants of the same animals analyzed in each generation to trace the altered DMCs/genes, which further control the variation in our study. Since there are multiple ways of power calculation in WGBS experimental design, and each result has different statistical power, we optimized our study based on previous studies, including ours, and used multiple ways to maximize our statistical power under current conditions.

3. The F3 generation showed higher birth weight (BWT) in the control group, while the F4 generation reversed this trend, with higher BWT in the methionine group. The article does not explain this phenomenon. Please provide an interpretation.

Response: We thank the reviewer for bringing this important issue to our attention. Several studies have shown that the maternal environment can influence offspring birth weight across generations; however, the underlying mechanisms remain unclear. A negative correlation between direct and maternal genetic effects has been previously reported in multiple livestock studies. For instance, genes associated with increased body weight in the dam may restrict fetal growth in her offspring (Yin and König, 2018). This pattern is consistent with evolutionary theories of parent–offspring conflict, which propose that optimal birth weight may differ between the interests of the mother and the fetus.

Additionally, DNA methylation marks are reversible across subsequent generations, a phenomenon observed in both our study and others (e.g., Braz et al., 2022). If the F0 paternal methionine supplementation induced epigenetic modifications that mimic or interact with such antagonistic maternal-fetal effects, this could explain the reduced birth weight observed in F3 (potentially reflecting inherited direct effects), followed by increased birth weight in F4 (possibly due to emerging or compensatory maternal effects). Finally, we note the presence of hundreds of de novo differentially methylated cytosines (DMCs), which may have further contributed to the generational reversal of phenotypic effects.

A summary of this discussion was added to the Discussion Section Lines 286-295

Yin, T. & König, S. Genetic parameters for body weight from birth to calving and associations between weights with test-day, health, and female fertility traits. Journal of Dairy Science, 2018, 101(3), 2158-2170.

Braz, C.U.; Taylor, T.; Namous, H.; Townsend, J.; Crenshaw, T.; Khatib, H. Paternal Diet Induces Transgenerational Epigenetic Inheritance of DNA Methylation Signatures and Phenotypes in Sheep Model. PNAS Nexus 2022, 1, doi:10.1093/pnasnexus/pgac040.

4. In Figure 1, key gene loci for the F3/F4 generations are not clearly labeled. Please supplement the figure or provide an explanation.

Response: Since adding gene names will make the figure overcrowded, we provided those genes as a supplementary file and added a sentence on line 234 pointing to this, as suggested by the reviewer.  

5. The distinction between "transgenerational epigenetic inheritance" and "intergenerational effects" is unclear. We suggest briefly explaining and standardizing the definitions of these terms.

Response: Thank you for bringing this to our attention. A clear definition showing the distinction between the two was added in lines 43-47, as suggested by the reviewer.

6. The discussion cites transgenerational studies in rats and mice but does not clarify the similarities and differences in epigenetic regulation between sheep and rodents. We recommend adding a species-specific analysis to avoid over-extrapolating conclusions.

Response: Epigenetic regulatory mechanisms are highly conserved across mammalian species, particularly the role of DNA methylation in controlling gene expression. This includes the conserved function and sequence of DNA methyltransferases (DNMTs), the enzymes responsible for adding methyl groups to specific cytosine residues within the genome (Moore et al., 2013; Ambrosi et al., 2017).

Moore, L., Le, T. & Fan, G. (2013). DNA Methylation and Its Basic Function. Neuropsychopharmacol 38, 23–38 (2013). https://doi.org/10.1038/npp.2012.112

Ambrosi, C., Manzo, M., Baubec, T. (2017). Dynamics and Context-Dependent Roles of DNA Methylation. J Mol Biol 429(10):1459-1475. doi: 10.1016/j.jmb.2017.02.008.

Reviewer 2 Report

Comments and Suggestions for Authors

This authors presented compelling evidence for transgenerational epigenetic inheritance (TEI) in a sheep model. Based on the previous work that demonstrated DNA methylation and phenotypic changes in the F1 and F2 generations following paternal methionine supplementation, the authors extended their analysis to F3 and F4 generations. By utilizing whole-genome bisulfite sequencing (WGBS) and phenotypic analysis, the authors identified differentially methylated cytosines (DMCs) and differentially methylated genes (DMGs) in sperm DNA, as well as growth and reproductive phenotypes. Notably, while few DMCs persisted across all generations, several DMGs, particularly 11, showed consistent methylation changes across five generations. The authors concluded that even a short-term dietary intervention in F0 males can induce long-lasting, heritable epigenetic and phenotypic changes.

  1. The introduction section provides a lot of useful information to help understand the concepts in the manuscript. However, there are still some points missing. Please add details explaining why methionine is particularly suited for investigating TEI, also please add the rationale why use sheep model instead of other models in this study.
  2. In the discussion section, please add a brief summary of  your previous research findings, like F0-F2, which can help readers to understand why you focus on F3-F4.
  3. In the manuscript, it shows a very significant association between DMCs/DMGs and phenotypic traits. However, this data is only based on bioinformatic analysis which lacks of causality. Functional validation, like gene knockdown test, could confirm if the observed methylation changes are directly responsible for the phenotypic effects.
  4. In the bioinformatic analysis, all DNA methylation data were derived from sperm. The phenotypic effects include somatic traits, such as body weight and muscle depth. Is it sufficient to connect sperm methylation data to specific somatic phenotypes? To be accurate, the methylation or gene expression data from relevant somatic tissues, like muscles, would be required for the conclusion.
  5. The authors highlighted that many TEI-DMGs are involved in growth and neural development, but there is limited discussion or references of whether these genes are known to be methylation-sensitive or involved in similar epigenetic regulation pathways in other models. Related references or detailed discussions would be required.
  6. Some traits, like birth weight, showed reversed effects in F3 vs. F4 generations. Do you have any data that correlates with specific methylation pattern shifts or with maternal/environmental covariates?

Thanks!

Author Response

Reviewer 2 comments and suggestions:

This authors presented compelling evidence for transgenerational epigenetic inheritance (TEI) in a sheep model. Based on the previous work that demonstrated DNA methylation and phenotypic changes in the F1 and F2 generations following paternal methionine supplementation, the authors extended their analysis to F3 and F4 generations. By utilizing whole-genome bisulfite sequencing (WGBS) and phenotypic analysis, the authors identified differentially methylated cytosines (DMCs) and differentially methylated genes (DMGs) in sperm DNA, as well as growth and reproductive phenotypes. Notably, while few DMCs persisted across all generations, several DMGs, particularly 11, showed consistent methylation changes across five generations. The authors concluded that even a short-term dietary intervention in F0 males can induce long-lasting, heritable epigenetic and phenotypic changes.

1. The introduction section provides a lot of useful information to help understand the concepts in the manuscript. However, there are still some points missing. Please add details explaining why methionine is particularly suited for investigating TEI, also please add the rationale why use sheep model instead of other models in this study.

Response: As suggested by the reviewer, we have added further information to clarify the rationale for using methionine and the sheep model. Methionine was selected because it functions as a methyl donor, and our previous studies have demonstrated its impact on sperm DNA methylation across generations (e.g., Gross et al., 2020; Braz et al., 2022; Townsend et al., 2023). Sheep were chosen as the model organism due to their relatively short generation interval compared to other large animals, a well-annotated genome, and—most importantly—their high twinning rate. The Polypay breed used in our experiments frequently produces twins and triplets, which is ideal for epigenetic studies, as one twin can be assigned to the treatment group and the other to the control group, allowing for tightly controlled comparisons.

We added this information in lines 106-113.

2. In the discussion section, please add a brief summary of your previous research findings, like F0-F2, which can help readers to understand why you focus on F3-F4.

Response: We highlighted the results of previous studies in lines 261-265 and pairwise comparisons with the current study across the discussion section.

3. In the manuscript, it shows a very significant association between DMCs/DMGs and phenotypic traits. However, this data is only based on bioinformatic analysis which lacks of causality. Functional validation, like gene knockdown test, could confirm if the observed methylation changes are directly responsible for the phenotypic effects.

Response: We thank the Reviewer for this comment. We also acknowledge the challenges of causality. In our previous study examining the F0–F2 generations, we demonstrated that the observed DNA methylation differences were correlated with gene expression levels (Braz et al., 2022). The current study extends this work by focusing on the transgenerational effects of paternal nutrition across five generations. We agree with the reviewer that future research should aim to investigate the causal relationships between phenotypic traits and the genes identified. Building on our current findings, future studies could explore the biological and functional roles of the TEI-associated differentially methylated genes (TEI-DMGs) that persist across four and five generations, with a particular focus on their contributions to phenotypic variation and mechanisms of transgenerational inheritance.

4. In the bioinformatic analysis, all DNA methylation data were derived from sperm. The phenotypic effects include somatic traits, such as body weight and muscle depth. Is it sufficient to connect sperm methylation data to specific somatic phenotypes? To be accurate, methylation or gene expression data from relevant somatic tissues, such as muscles, would be required to support the conclusion.

Response: Thank you for this valuable comment. We acknowledge that this is a complex and challenging issue. However, we consider sperm or oocytes to be the primary mediators of transgenerational epigenetic inheritance, as well as key regulators of embryonic development that can lead to phenotypic differences in offspring. In this study, we demonstrated stable, gene-level methylation differences and associated phenotypic changes in sperm across five generations. Notably, our previous work (Townsend et al., 2023) demonstrated that alterations in sperm DNA methylation were correlated with gene expression in the subsequent generation of embryos. Furthermore, a separate study by Braunschweig et al. (2012) linked diet-induced epigenetic changes transmitted through the male line to differences in gene expression and carcass traits in somatic tissues of pigs over two generations.

Townsend, J.; Braz, C.U.; Taylor, T.; Khatib, H. Effects of paternal methionine supplementation on sperm DNA methylation and embryo transcriptome in sheep. Environmental Epigenetics 2023, 9, dvac029, doi:10.1093/eep/dvac029.

Braunschweig, M.; Jagannathan, V.; Gutzwiller A.; Bee, G. Investigations on transgenerational epigenetic response down the male line in F2 pigs. PLoS One. 2012;7(2):e30583. doi: 10.1371/journal.pone.0030583.

5. The authors highlighted that many TEI-DMGs are involved in growth and neural development, but there is limited discussion or references of whether these genes are known to be methylation-sensitive or involved in similar epigenetic regulation pathways in other models. Related references or detailed discussions would be required.

Response: Indeed, our initial approach was to search for studies that directly linked methylation differences in these specific genes to the phenotypes observed. However, we were unable to find any published literature that identified these genes in connection with the traits in question. As an alternative, we conducted a thorough review of the literature and relevant databases to functionally annotate these genes in terms of their associated pathways and biological processes, as detailed in the Results and Discussion sections.

6. Some traits, like birth weight, showed reversed effects in F3 vs. F4 generations. Do you have any data that correlates with specific methylation pattern shifts or with maternal/environmental covariates?

Response: We thank the reviewer for bringing this important issue to our attention. Several studies have shown that the maternal environment can influence offspring birth weight across generations; however, the underlying mechanisms remain unclear. A negative correlation between direct and maternal genetic effects has been previously reported in multiple livestock studies. For instance, genes associated with increased body weight in the dam may restrict fetal growth in her offspring (Yin and König, 2018). This pattern is consistent with evolutionary theories of parent–offspring conflict, which propose that optimal birth weight may differ between the interests of the mother and the fetus.

Additionally, DNA methylation marks are reversible, a phenomenon observed in both our study and others (e.g., Braz et al., 2022). If the F0 paternal methionine supplementation induced epigenetic modifications that mimic or interact with such antagonistic maternal-fetal effects, this could explain the reduced birth weight observed in F3 (potentially reflecting inherited direct effects), followed by increased birth weight in F4 (possibly due to emerging or compensatory maternal effects). Finally, we note the presence of hundreds of de novo differentially methylated cytosines (DMCs), which may have further contributed to the generational reversal of phenotypic effects.

A summary of this discussion was added to the Discussion Section Lines 286-295.

Yin, T. & König, S. Genetic parameters for body weight from birth to calving and associations between weights with test-day, health, and female fertility traits. Journal of Dairy Science, 2018, 101(3), 2158-2170.

Braz, C.U.; Taylor, T.; Namous, H.; Townsend, J.; Crenshaw, T.; Khatib, H. Paternal Diet Induces Transgenerational Epigenetic Inheritance of DNA Methylation Signatures and Phenotypes in Sheep Model. PNAS Nexus 2022, 1, doi:10.1093/pnasnexus/pgac040.

Round 2

Reviewer 1 Report

Comments and Suggestions for Authors

The article has been greatly improved, it is recommended to accept it

Author Response

We thank the Reviewer for their constructive comments.